# Confirmative biophilic framework for heritage management

Hung-Ming Tu  *

Department of Horticulture, National Chung Hsing University, Taichung, Taiwan

* hmtu@dragon.nchu.edu.tw

## Abstract

Heritage managers often reuse heritage sites to attract tourists and conserve the sites. Not all adaptive heritage reuses achieve sustainable development. Biophilia is an innate, biological tendency to be close to natural and cultural elements, which may be a critical motivation for achieving sustainable heritage management. Past studies used qualitative and quantitative methods to extract participants' pull and push motivations toward heritage tourism and captured the six motivations of the biophilia framework that should be confirmed: heritage architecture; art activity; wide nature; regional attraction; recreational benefits; and long-term values. The purpose of this study was to conduct a confirmatory factor analysis to test the biophilia framework for understanding biophilic heritage tourism. A questionnaire with 18 items of heritage tourism motivation was used to explore the purpose of this study. A total of 193 valid questionnaires were obtained. Confirmatory factor analysis was used to examine the six motivations of the biophilia framework. The results indicated that heritage tourism motivation consisted of a second-order six-factor structure with high validity and reliability. These six dimensions reflected the biophilic requirements and provided a biophilic planning principle to achieve sustainable heritage management to satisfy the needs of the participants.

## Introduction

Heritage preservation plays a critical role in preserving the human past, present, and future [1]. Heritage architecture is one of the major subjects in heritage preservation because of its historical meaning [2]. The preservation of architecture produces an emotional connection among people, places, and traditions, and nurtures human spirit and life [3]. For sustainable development, adaptive reuse is an appropriate method to change the value and function of architecture while satisfying the needs of new participants [4,5]. Heritage managers often reuse heritage sites to conserve heritage sites and attract tourists to promote tourism [6–8]. Heritage tourism brings economic contribution [9]. However, not all adaptive heritage reuses achieve sustainable development because many fail to satisfy the new participants' needs and attract travellers [10]. Some heritage cases with high cultural values produce low tourism and experiential values due to isolation from other attractions, small scale, low uniqueness, and poor environment [11]. Improving tourism opportunities in heritage sites can promote

Research Ethics Committee (approval number: NCKU HREC-E-107-119-2). Data are available from the National Cheng Kung University Human Research Ethics Committee (contact via em51020@email.ncku.edu.tw) for researchers who meet the criteria for access to confidential data.

**Funding:** The author is sincerely grateful to the Ministry of Science and Technology (MOST) in Taiwan for its full support (Project No. MOST 107-2410-H-005-028-MY3). The funders had no role in study design, data collection and analysis, decision to publish, or preparation of the manuscript.

**Competing interests:** The authors have declared that no competing interests exist.

economic benefits [12]. Achieving sustainable heritage management has become an essential and practical as well as an academic issue.

The biophilic principle may be a critical consideration for achieving sustainable heritage management. Biophilia is a biological and an innate tendency to be close to natural and cultural elements related to human evolution processes, such as food, safety, and security [13–15]. Biophilic elements satisfy the needs of health and well-being and promote emotional connection to induce a place's cultural identity [3,14,16]. Biophilic considerations may satisfy participants' needs and induce cultural identity in heritage tourism to preserve heritage. Biophilia may play a vital role as a background in heritage tourism. However, past references of biophilia have focused on natural elements [13,14]. Few references have connected the relationship between heritage and biophilia.

Tu used qualitative and quantitative methods to extract participants' pull and push motivations toward heritage tourism [10,17]. They extracted six dimensions through exploratory factor analysis, including heritage architecture, art activity, wide nature, regional attraction, recreational benefits, and long-term values. These six dimensions were the common considerations for visiting heritage sites. The wide nature dimension represents the core elements of biophilia. The heritage architecture, art activity, and regional attraction, reflect the familiarity of evolutionary elements to increase biophilic connections, which satisfy biophilic requirements and become important heritage tourism considerations. The recreational benefits and long-term values reflect the leisure needs of heritage tourism. These findings show that heritage architecture, wide nature, and recreational benefits were critical considerations, reflecting the possibility of biophilic leisure in heritage tourism. This framework of six dimensions should be confirmed, although Tu's studies provide preliminary evidence for the possibility of biophilic leisure in heritage tourism. In order to fill the gap, the purpose of this study was to conduct a confirmatory factor analysis (CFA) to test Tu's structure of participants' motivation for understanding biophilic leisure in heritage tourism. This study provides biophilic principles for sustainable heritage management.

## Literature review

### Biophilia and heritage preservation

Biophilia is a biological tendency whereby human beings need to be connected to nature associated with their health and well-being [14,16]. Human evolution is biologically developed in an adaptive response to nature and an adaptive response to the urban environment [14]. This adaptive response stress and health problems occur in urban environments [18]. In contrast, natural elements, designs, and patterns often elicit consistent aesthetic and preference responses under various cultural and geographical circumstances [13]. This condition reflects the dependent relationships with food, safety, and security from natural resources for survival and reproduction in human history [13,15]. Nature often produces happiness, which can be explained by a feeling of security and familiarity [19]. Therefore, several practitioners use biophilic design to improve well-being and happiness in urban buildings and the environment [16].

The concept of biophilia tried to link cultural and ecological attachment to places. Humans have a biological tendency to connect with familiar and cultural places, reflecting that territorial inclination comes from the control of resources, safety, security, and mobility [14]. Familiarity and cultural places produce an ecological connection and emotional attachment to a place called the spirit of place, promoting human and cultural identity to conserve place [3,14]. Therefore, biophilia is the foundation of cultural identity, which also produces a healing effect

in traditional places [20]. Therefore, biophilia has a biological tendency to connect with not only nature but also cultural places.

Nature also provides substantial cultural and social value to humans [15], although few studies have pointed to the relationship between nature and culture. Nature experiences produce cultural benefits, such as environmental concerns and attachment [13]. Biophilic natural elements, which often appear in the environment of leisure, vacation, or honeymoon rather than the daily environment, form an essential social background to relaxing memories and contribute to life satisfaction and cultural benefits [15]. One study of heritage cities captured urban architecture and green spaces as a common factor in attracting tourists [21]. The loss of nature potentially leads to the loss of memory and reduces cultural values [15]. Nature plays an essential role in leisure and travel time. Therefore, nature encourages an emotional attachment to familiar places, which further constructs biophilic value and conserves place and heritage [3,14]. The integration of culture with nature produces a unique cultural identity [3].

## Biophilic motivation toward heritage tourism

Few studies have explored the relationship between biophilia and heritage. Previous studies indicated that participants' motivations for heritage tourism covered six dimensions: heritage architecture, art activity, wide nature, regional attraction, recreational benefits, and long-term values [17]. These six dimensions can be linked to the concept of biophilia. The dimension of wide nature is the core link that explains the relationship between heritage and biophilia. Nature is an important element of the aesthetic response to present feelings of harmony [13]. The physical nature of the heritage environment is an important consideration for heritage tourism [22]. The wide space and view reflect the fact that people prefer a wide vista to perceive dangers and obtain safety and security [14]. Therefore, a broad view of nature, "wide nature," provokes positive feelings in urban and leisure environments. Some studies have indicated that culture should positively merge with nature to produce health or positive connections in the vernacular landscape [3].

The natural light, bright colors, rich borders, frames, moldings, ornaments, natural materials, balanced curves, water, plants, and non-threatening animals are important considerations for a biophilic design in the built environment [23–25]. In the heritage architecture dimension, natural fractal patterns may play an important role in visual feeling. A natural fractal pattern is a special type of complexity that produces a positive effect on aesthetic experience, preference, emotion, and stress reduction [26]. This is because our ancestors may have identified the non-fractal patterns of animals to avoid dangerous situations and thereby adopted a sensitivity toward natural fractal patterns [26]. In a built environment, the architecture, flooring design, windows, and decorations also use natural fractal patterns to promote an aesthetic experience and preference [26]. For example, the representations of plants, animals, and people in the built environment through photographs, paintings, or sculptures contribute to biophilia [23,24]. The designs of several classical architecture ornaments are derived from leaves, flowers, and animal skins [27]. However, modern architecture often portrays minimalism to erase biophilic features [23,24], which may lead to a reduction in an individual's positive reflection on preference and emotion.

Vernacular architecture often presents the richness and variety of biophilic features to produce a healing effect due to the complex geometry of the neurobiological system's preferences [23]. The geometric pattern of traditional and historical architecture connects people and places to satisfy human needs and produces a spirit of place and a healing effect through an evolved identity from a biophilic foundation [20]. Therefore, heritage architecture is a biophilic geometry pattern that links the relationship between architecture and identity. Modern

architecture may lack biophilic linkages and produce environmental stress. Part of the traditional architecture imitates nature to produce biophilic environments [20]. For example, a Japanese-style heritage often provides a large space for natural elements or merges with the natural environment to meet participants' preferences [10]. The human body can observe differences in nonbiophilic features, inducing hostility and reducing the healing effect when they appear in traditional architecture [20]. From the above explanation, it follows that heritage architecture reflects a core motivation of linking biophilia through the spirit of place and satisfaction of intrinsic needs.

Art activity is an important attraction in heritage tourism. Cultural events, festivals, and experiential consumption are important attractions for producing positive emotions in heritage tourism [28–32]. Interactive elements, themes, and designs are important factors in learning and entertainment [33]. Few studies have explained the relationship between art activity and biophilia. Art activity is also a cultural activity that increases the link between cultural attachment and a biophilic foundation. The direct explanation is that leisure activity is a kind of biophilic leisure associated with the presentation of a natural background, reflecting opportunities arising from social activities and improving social relationships [15,34].

The dimension of regional attraction focuses on recreational attractions, historical streets, and cultural heritage. Consequently, the recreational attractions, historical streets, and cultural heritage also reflect the needs of the biophilic foundation. Previous studies have suggested that biophilic design should construct coherent connections on a regional scale [14], such as coherent natural elements, cultural elements, and biophilic leisure. The coherent connections also reflect the high transportation connectivity, which leads to greater opportunity to be close to biophilic spaces [35].

Promoting life quality is one of major motivations for participating tourism [36]. The dimension of recreational benefits and long-term values is the recreational push factor that reflects intrinsic motivation, which should be satisfied in heritage tourism [17]. The heritage architecture, art activity, wide nature, and regional attraction may satisfy the participants' recreational benefits and long-term values to become important pull factors. For example, natural experience promotes relaxation and calm, sharpens vitality and awareness, improves physical fitness, and enhances creativity [13].

## Methods

This study's research ethics were assessed and approved by the National Cheng Kung University Human Research Ethics Committee (approval number: NCKU HREC-E-107-119-2).

### Instrument

The questionnaire was used to explore the purpose of this study, including personal background, heritage tourism experience, and important considerations for heritage tourism. Personal background included gender, age, marital status, education level, monthly income excluding fixed expenses, and occupation. The experience of heritage tourism was used to understand the frequency of visiting heritage sites in the last year and travel partners. The 7-point scale was used to assess the frequency of visiting heritage sites in the last one year from 1 = very infrequently to 7 = very frequently. The multiple-choice question of travel partners comprised own, family, friends, and others.

Important considerations for heritage tourism were captured from the studies of Tu [10,17]. Tu captured the heritage pull-push factors through open-ended interviews of visitors in the cases of adaptive heritage reuse and generalised seven pull-push factors: heritage, activity, natural environment, regional environment, self-growth, health benefits, social benefits,

and cultural benefits [10], based on which Tu generated 56 initial items of the pull-push factor toward heritage tourism and used exploratory factor analysis to extract the common dimensions [17]. Tu extracted six pull-push dimensions and 24 items, including heritage architecture, art activity, wide nature, regional attraction, recreational benefits, and long-term values [17]. The Cronbach's α of each extracted dimension was 0.83 to 0.90, indicating high reliability. Some dimensions showed lower factor loading because the criteria of factor loading are not rigorous in exploratory factor analysis. The lower factor loading probably affects the results of the confirmatory factor analysis. This study extracted three items with higher factor loadings per dimension to form a parsimonious model with 18 items from Tu [17] to test the biophilia framework. Table 1 presents the English language translations of the initial items and six dimensions from the original Chinese items.

Previous studies used the important level to capture the push and pull factors [17,30]. Therefore, the respondents were asked to score the importance level on a 7-point scale from 1 = very unimportant to 7 = very important of each item when visiting a heritage site. Tu provided a definition of heritage and a list of well-known heritage cases in Taiwan to help participants understand the concept of heritage and better assess the study items because some respondents did not understand the definition or cases of heritage [17].

## Data collection

This study collected data from June 7 to 8, 2019, at the Calligraphy Greenway in Taichung City in Central Taiwan. Calligraphy Greenway is a major recreational metropolitan park and the

**Table 1. Initial heritage tourism items.**

| Dimensions and items |
| --- |
| A: Heritage architecture |
| A1: Heritage architecture itself. |
| A2: Style of heritage architecture. |
| A3: Beautification of heritage architecture. |
| B: Art activity |
| B1: The heritage often holds dynamic art activities. |
| B2: The heritage often holds static art activities. |
| B3: The heritage often holds holiday markets. |
| C: Wide nature |
| C1: The heritage's outdoor environment has several natural elements. |
| C2: You can view the natural landscape. |
| C3: The heritage's outdoor environment is wide. |
| D: Regional attraction |
| D1: The heritage's surrounding region has historical streets. |
| D2: The heritage's surrounding region has other cultural heritages. |
| D3: The heritage's surrounding region has other recreational attractions. |
| E: Recreational benefits |
| E1: Heritage tourism can provide a novel and fun experience. |
| E2: You can share travel experiences with family and friends. |
| E3: Heritage tourism can promote interaction with family and friends. |
| F: Long-term values |
| F1: Heritage tourism can provide a sense of achievement. |
| F2: Heritage tourism can bring a good life. |
| F3: Heritage tourism can improve health. |

most notable greenbelt around Taichung City, popular with all age groups. Several local studies selected Calligraphy Greenway to collect tourist data [37]. This study surveyed subjects who were over 20 years old at the four major recreational areas of Calligraphy Greenway by convenience sampling. Participants were informed of the purpose, content, rights, and rewards of this study through a questionnaire participant information sheet. The reward for this study was a gift after the completion of the questionnaire.

In the determination of sample size, confirmatory factor analysis (CFA) should consider the number of factors, the number of indicators, the magnitude of factor loadings, and the magnitude of factor correlations to determine statistical power [38]. According to the number of factors and indicators in this study (Table 1), high factor loadings (0.70), and high factor correlations (0.50), the acceptable minimum sample size was 150 from the simulations of Wolf et al. [38]. Therefore, this study distributed 200 questionnaires. A total of 193 valid questionnaires was obtained, excluding missing values. The effective response rate was thus 96.5 per cent.

The proportion of women (51.3%) was close to that of men (48.7%) (Table 2). Most subjects were aged 20 to 39 years (66.4%). Half of the participants were married (52.8%). Four-fifths of the subjects had higher education (86.5%), including college, university, and postgraduate education. Half of the subjects' monthly income was lower than NT$ 20,000 (equal to US$ 625) (50.5%). The subjects had a larger proportion of blue-collar occupations (18.7%) and professionals (18.7%). The previous year's visiting heritage frequency was ordinary (mean and SD were 3.32 and 1.35). The major travel partners were family (63.2%) and friends (25.9%). The proportion of each background variable was similar to that of Tu [17].

## Data analysis

Confirmatory factor analysis (CFA) is a useful method for providing a confirmatory test of measurement theory [39]. CFA was used to analyze the factor structure of heritage tourism motivations. Some theoretical contexts may specify a structure that explains the relationships among the factors, called a second-order factor model [40]. The second-order factor model means that all factors are related to a higher-order, meaningful, and interpretable factor [41]. This study hypothesised that the biophilia hypothesis was a second-order factor construct to explain the relationships between heritage tourism motivations. CFA has the function of examining the second-order factor model of psychological construct and comparing the variations of different models through competing processes. Therefore, this study first examined the first-order factor model of heritage tourism motivations and the correlations of six dimensions. We further measured the second-order factor model to examine meaningful and interpretable constructs based on the biophilia hypothesis.

Harman's single-factor test was used to test common method variance (CMV) using SPSS software [42,43]. Maximum likelihood estimation (ML) was used to implement the CFA using AMOS software. First, univariate skewness, univariate kurtosis, and multivariate kurtosis were tested to ensure normality. Multivariate kurtosis should be lower than $p^*(p + 2)$, where $p$ is the number of variables [44]. Therefore, multivariate kurtosis should be lower than 360 to ensure multivariate normality. Second, the validity of the measurement model was tested. The suggested values of goodness-of-fits were determined from the past references: (1) Chi-square ($\chi^2$)/degrees of freedom ($df$) should be lower than 3.0; (2) the comparative-fit index (CFI) should be greater than 0.90; (3) the standardised root mean square residual (SRMR) should be lower than 0.10; (4) the root mean square error of approximation (RMSEA) should be between 0.03 and 0.08; and (5) the parsimonious normed-fit index (PNFI) should be relatively high when comparing models [39]. The suggested consideration of convergent validity and

**Table 2. The personal backgrounds of study subjects.**

| Personal background | N | % |
|---|---|---|
| Gender | | |
| Male | 94 | 48.7 |
| Female | 99 | 51.3 |
| Age (years) | | |
| 20–29 | 82 | 42.5 |
| 30–39 | 46 | 23.9 |
| 40–49 | 29 | 15.0 |
| 50–59 | 27 | 14.0 |
| 60 or older | 9 | 4.6 |
| Marital status | | |
| Single | 87 | 45.1 |
| Married with no children | 8 | 4.1 |
| Married with children | 94 | 48.7 |
| Other | 4 | 2.1 |
| Education level | | |
| Primary | 1 | 0.5 |
| High school | 25 | 13.0 |
| College | 23 | 11.9 |
| University | 107 | 55.4 |
| Postgraduate | 37 | 19.2 |
| Monthly income (excluding fixed expenses) | | |
| Less than NT $ 10,000 | 53 | 27.5 |
| NT $ 10,001–20,000 | 45 | 23.3 |
| NT $ 20,001–30,000 | 35 | 18.1 |
| NT $ 30,001–40,000 | 21 | 10.8 |
| NT $ 40,001–50,000 | 15 | 7.8 |
| More than NT $ 50,001 | 24 | 12.4 |
| Occupation | | |
| Administrator | 13 | 6.7 |
| Professional | 36 | 18.7 |
| Technician | 20 | 10.4 |
| Clerk | 25 | 13.0 |
| Blue collar occupation | 35 | 18.2 |
| Housewife | 17 | 8.8 |
| Retirement | 9 | 4.7 |
| Other | 38 | 19.7 |
| Frequency of visiting heritage last year | | |
| Very infrequent | 24 | 12.4 |
| Infrequent | 30 | 15.5 |
| Somewhat infrequent | 41 | 21.2 |
| Ordinary | 67 | 34.7 |
| Somewhat frequent | 23 | 11.9 |
| Frequent | 6 | 3.1 |
| Very frequent | 2 | 1.0 |
| Travel partner | | |
| Own | 13 | 6.7 |
| Family | 122 | 63.2 |

(*Continued*)

**Table 2.** (Continued)

| Personal background | N | % |
|---|---|---|
| Friend | 50 | 25.9 |
| Other | 8 | 4.1 |

discriminant validity were determined: (1) all factor loadings should ideally be higher than 0.70; (2) average variance extracted (AVE) should be higher than 0.50; (3) composite reliability (CR) should be higher than 0.70; and (4) the AVE value should be greater than the squared correlation estimate [39]. The target coefficient is the ratio of $\chi^2$ between the first-order and second-order models and should be close to 1.00 when comparing the first- and second-order models, which means first-order factors can be accounted for by the second-order construct [45].

# Results

## The test of the first-order construct

In the CMV test, the first unrotated factor captured 33.3 per cent of the variance in the data, which means that CMV did not produce a problem in our data. In the test of normality, univariate skewness (range: −0.17 to −1.03), univariate kurtosis (range: −0.75 to 1.86), and multivariate kurtosis (92.3) showed acceptable values. The initial test of the 18 items of a first-order six-factor model produced sufficient goodness of fit ($\chi^2/df$ = 1.73, CFI = 0.90, SRMR = 0.05, RMSEA = 0.06, PNFI = 0.70) (Table 3 and Fig 1). The dimensions of heritage architecture, art activity, wide nature, recreational benefits, and long-term values showed acceptable factor loading (range: 0.70 to 0.94), CR values (range: 0.81 to 0.88), and AVE values (range: 0.58 to 0.70). One item of regional attraction showed a lower factor loading (0.59) and induced lower AVE values (0.49) of the dimension of regional attraction, which was still an acceptable value according to Hair et al. [39]. Therefore, the CFA process retained the item with lower factor loading. All AVE values were greater than the squared correlation estimates (Table 4). Overall, the above test's outcomes indicated that all six dimensions were valid and reliable in presenting heritage tourism motivations.

## The test of second-order construct

The higher-order factor model was tested using the 18 items of a second-order six-factor model (Fig 2). Model comparisons are presented in Table 3. These two models performed similar goodness-of-fit measures. The second-order six-factor model showed adequate goodness of fit ($\chi^2/df$ = 1.93, CFI = 0.89, SRMR = 0.07, RMSEA = 0.07, PNFI = 0.73), although the

**Table 3. Model comparisons of heritage tourism.**

| Model Fit | Initial first-order six-factor model | Revised second-order six-factor model |
|---|---|---|
| Degrees of freedom (df) | 120 | 129 |
| Chi-square($\chi2$) | 207.52 | 247.83 |
| $\chi2/df$ | 1.73 | 1.92 |
| CFI | 0.90 | 0.89 |
| SRMR | 0.05 | 0.07 |
| RMSEA | 0.06 | 0.07 |
| PNFI | 0.70 | 0.73 |

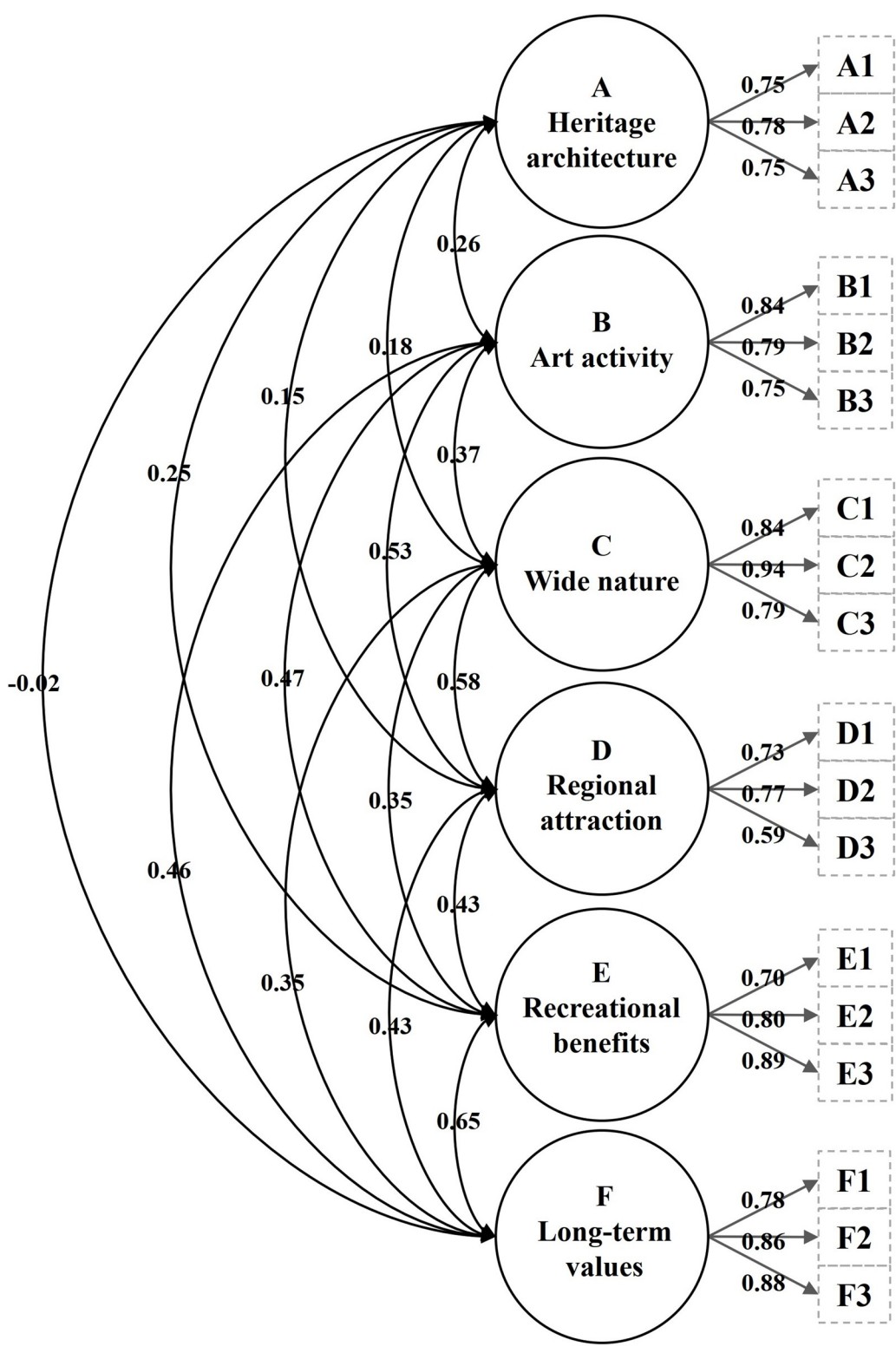

**Fig 1. Initial first-order six-factor model of heritage tourism motivation.**

**Table 4. Discriminant validity between the six dimensions of heritage tourism from the initial first-order six-factor model.**

|  | A | B | C | D | E | F |
|---|---|---|---|---|---|---|
| A: Heritage architecture | 0.58[a] |  |  |  |  |  |
| B: Art activity | 0.03[b] | 0.64[a] |  |  |  |  |
| C: Wide nature | 0.07[b] | 0.14[b] | 0.70[a] |  |  |  |
| D: Regional attraction | 0.02[b] | 0.34[b] | 0.29[b] | 0.49[a] |  |  |
| E: Recreational benefits | 0.06[b] | 0.12[b] | 0.22[b] | 0.18[b] | 0.64[a] |  |
| F: Long-term values | 0.00[b] | 0.12[b] | 0.21[b] | 0.19[b] | 0.42[b] | 0.70[a] |

[a] Average variance extracted (AVE).

[b] The square of the correlation estimate between two dimensions.

goodness of fit of the first-order model was better than that of the second-order model, except for PNFI. The higher-order model's goodness-of-fit must be lower than that of the first-order model [45]. All the dimension factor loadings in the sub-constructs, CR values, and AVE values were similar to the first-order model results (Table 5).

In the main construct, the heritage architecture dimension produced a low factor loading (0.24), and the AVE value (0.40) was lower than the suggested value (Table 5). The CR value (0.79) was higher than the suggested value, which means that the construct's convergent validity was adequate, although it had lower AVE [46,47]. Despite the low factor loading, it was retained because of the adequate CR value.

Comparing the two models, the target coefficient of 0.84 was close to 1.00, providing reasonable evidence of a second-order model, meaning that the second-order model can explain 84 per cent of the variation among the six dimensions of the first-order model. The second-order model was well-supported and indicated that the factor structure of motivations toward heritage tourism consists of six dimensions and integrates a meaningful second-order structure that covers all dimensions.

## Discussion

### Six dimensions of heritage tourism motivation

This study is the first to provide a confirmative framework for biophilia heritage tourism and indicates that heritage tourism motivation consists of a second-order six-factor structure with high validity and reliability through the CFA process. The six dimensions were heritage architecture, art activity, wide nature, regional attraction, recreational benefits, and long-term values. The second-order structure indicated that the six dimensions were integrated into a common high-order factor, which reflected the biophilic and recreational requirements in heritage tourism from the biophilia hypothesis. Visiting heritage experiences allows biophilic recreation to satisfy biophilic and recreational requirements through heritage architecture, art activity, wide nature, and regional attraction. The results of this study provide a biophilic planning principle to achieve sustainable heritage management.

Previous studies indicated that tangible heritage architecture (e.g., original buildings, decoration of buildings, and indoor design) and intangible culture (e.g., art, stories, legends, and traditional appliances) both affect participants' perceptions of heritage sites [48–50]. This study further showed that the dimensions of heritage architecture and art activity are both important tangible and intangible elements that induce the decision to visit heritage. The heritage architecture dimension indicated that heritage architecture, beautification of heritage architecture, and heritage architecture style were critical considerations. The elements of

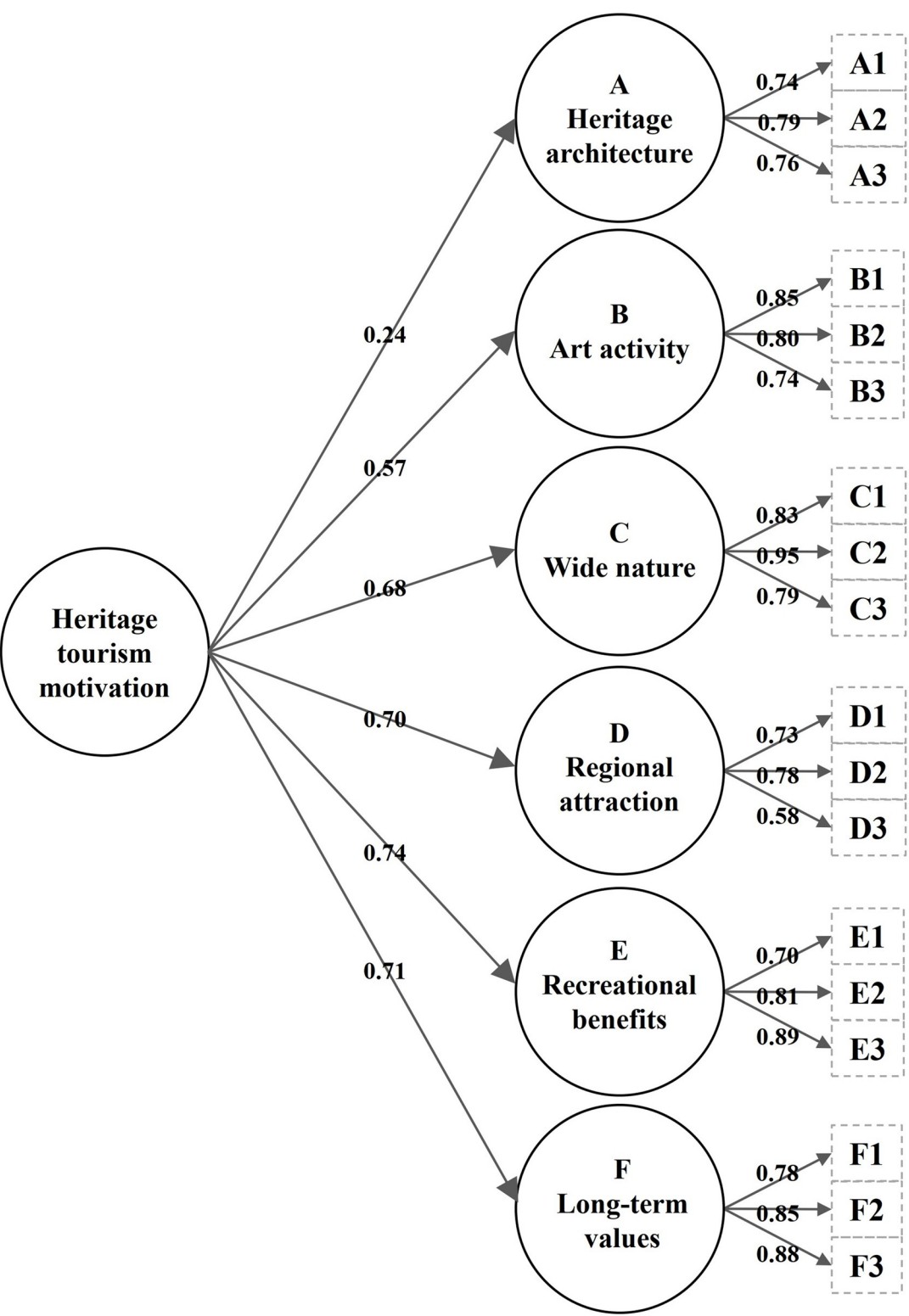

**Fig 2. Revised second-order six-factor model of heritage tourism motivation.**

**Table 5. Confirmatory factor analysis of the 18 items of the revised second-order six-factor model.**

| Construct, dimensions and items | Mean (S.D.) | Factor loading | t value | R² |
|---|---|---|---|---|
| Main construct (CR = 0.79, AVE = 0.40) | | | | |
| A: Heritage architecture | 5.80(1.04) | 0.24 | | 0.06 |
| B: Art activity | 4.60(1.43) | 0.57 | 2.54 | 0.46 |
| C: Wide nature | 5.34(1.37) | 0.68 | 2.48 | 0.33 |
| D: Regional attraction | 5.01(1.33) | 0.70 | 2.50 | 0.49 |
| E: Recreational benefits | 5.52(1.21) | 0.74 | 2.52 | 0.54 |
| F: Long-term values | 4.68(1.56) | 0.71 | 2.54 | 0.50 |
| Sub construct | | | | |
| A: Heritage architecture (CR = 0.81, AVE = 0.58) | | | | |
| A1: Heritage architecture itself. | 5.91(1.00) | 0.74 | | 0.55 |
| A2: Style of heritage architecture. | 5.83(1.04) | 0.79 | 8.84 | 0.62 |
| A3: Beautification of heritage architecture. | 5.67(1.07) | 0.76 | 8.78 | 0.57 |
| B: Art activity (CR = 0.84, AVE = 0.64) | | | | |
| B1: The heritage often holds dynamic art activities. | 4.53(1.46) | 0.85 | | 0.73 |
| B2: The heritage often holds static art activities. | 4.63(1.35) | 0.80 | 11.04 | 0.63 |
| B3: The heritage often holds holiday markets. | 4.64(1.48) | 0.74 | 10.40 | 0.55 |
| C: Wide nature (CR = 0.92, AVE = 0.70) | | | | |
| C1: The heritage's outdoor environment has several natural elements. | 5.42(1.36) | 0.83 | | 0.70 |
| C2: You can view the natural landscape. | 5.26(1.40) | 0.95 | 15.45 | 0.89 |
| : The heritage's outdoor environment is wide. | 5.33(1.34) | 0.79 | 12.79 | 0.62 |
| D: Regional attraction (CR = 0.74, AVE = 0.49) | | | | |
| D1: The heritage's surrounding region has historical streets. | 5.22(1.23) | 0.73 | | 0.54 |
| D2: The heritage's surrounding region has other cultural heritages. | 4.85(1.38) | 0.78 | 8.12 | 0.60 |
| D3: The heritage's surrounding region has other recreational attractions. | 4.95(1.39) | 0.58 | 6.83 | 0.34 |
| E: Recreational benefits (CR = 0.84, AVE = 0.64) | | | | |
| E1: Heritage tourism can provide a novel and fun experience. | 5.72(1.15) | 0.70 | | 0.49 |
| E2: You can share travel experiences with family and friends. | 5.55(1.21) | 0.81 | 9.95 | 0.65 |
| E3: Heritage tourism can promote interaction with family and friends. | 5.30(1.28) | 0.89 | 10.36 | 0.79 |
| F: Long-term values (CR = 0.88, AVE = 0.70) | | | | |
| F1: Heritage tourism can provide a sense of achievement. | 4.45(1.51) | 0.78 | | 0.62 |
| F2: Heritage tourism can bring a good life. | 4.91(1.52) | 0.85 | 12.44 | 0.73 |
| F3: Heritage tourism can improve health. | 4.68(1.64) | 0.88 | 12.75 | 0.78 |
| Model Fit | | | | |
| Degrees of freedom (df) | 129 | | | |
| Chi-square(χ2)/df | 1.92 | | | |
| Comparative-fit index (CFI) | 0.93 | | | |
| Standardised root mean square residual (SRMR) | 0.07 | | | |
| Root mean square error of approximation (RMSEA) | 0.07 | | | |
| Parsimonious normed-fit index (PNFI) | 0.73 | | | |

Note. CR = composite reliability; AVE = average variance extracted.

architecture and its beautification are consistent with the considerations of biophilic design. Natural fractal patterns and representations of plants, leaves, flowers, animals, and people should be considered to promote biophilic experiences and health [26]. Japanese style and beautification are the favorite elements in Taiwan's study [10]. The reuse of Taiwan's Japanese-style heritage preserves the architecture and the historical neighborhood, large gardens,

old trees, and a variety of plants in a positive role in the regional environment [51]. Traditional Japanese gardens present small-scale biophilic interventions in people's doorways [27]. The biophilic planning principle matches the reuse cases of Japanese heritage.

The social and cultural interactions between tourists and residents produce emotional engagement in heritage tourism [52]. Cultural events, festivals, and gastronomy are common social activities to produce positive emotion [28–30,32] and induce place attachment and intentions of further revisits [53]. This study indicated that the dimension of art activity indicated that dynamic art activities, static art activities, and holiday markets were critical intangible cultural elements. These activities may induce emotional engagement in tourism. Heritage tourism should consider activity programming to attract travellers and achieve economic benefits. In recent years, holding a holiday market has become a common and successful method to attract young travellers and tourists. The holiday market covers not only dynamic and static art activities, but also the experience of gastronomy. Several studies have pointed out that gastronomy satisfies specific tourists for enjoying cultural food [54,55]. According to the biophilia concept, dynamic and static art activities and holiday markets are intangible cultural elements that connect past life self-experience and evolutionary elements and may induce place attachment to close heritage. This type of activity extends the broad definition of biophilia.

Heritage and landscape have common topics and territories [56]. Although the natural environment is an essential factor in heritage tourism [48,57–59], its natural environment can easily be ignored in the process of adaptive reuse. The tourists prioritise the rich natural environments [60] and produce important destination image [61]. This study indicated that the dimension of wide nature was a determinant dimension of tourism as a form of biophilia heritage. The three items indicated that natural elements, natural landscapes, and wide landscapes were important values in the heritage's outdoor environment. The heritage's natural environment produces positive emotions, creates activities, and connects people to history and culture [58,62]. Heritage sites should plan wide natural landscapes. Heritage items without greenspace opportunities may easily fail to successfully adapt to reuse, except for their historical meaning. Moreover, the purpose of heritage tourism may affect the needs of the natural environment. Celebration activities reduce natural needs because social activities or traditional ceremonies need high accessibility and thus prioritise biophilic needs [15]. However, weddings, honeymoons, vacations, and fun activities often occur in a natural environment [15]. Nature is an important background factor that attracts travelers with different purposes in heritage tourism.

Considering their effectiveness, cost, and benefit-cost ratio, Xue et al. indicated that natural window views, natural ventilation, and natural landscape promotion with minimal management are the three most essential strategies in a built environment from the perspective of stakeholders [63]. This study also supported the proposition that heritage architecture sites should consider natural window views and natural landscape promotion. One study indicated that the natural environment should not obstruct the view of historical structures at the cost of promoting landscape preferences [64], implying that the aesthetic and natural view of historical structures should be an important consideration while undertaking heritage greening activities. Interestingly, natural decoration and ornamentation are not considered cost-effective or favored in the general built environment by stakeholders [63]. This study suggested that heritage decoration and ornamentation are important considerations for heritage tourism. Therefore, the decoration of natural fractal patterns should be considered for heritage protection or tourism.

Biophilic design considers multiple scales to connect biophilic elements from interior spaces, architecture, and surrounding landscapes to urban and regional scales [14]. The results of this study indicate that the dimension of regional attraction indicates that historical streets,

other cultural heritage sites, and recreational attractions in the heritage's surrounding region are important considerations in heritage tourism. The selection of adaptive heritage reuse should consider regional historical streets, cultural heritage, and recreational attractions. These regional cultural and recreational attractions reflected that the biophilic planning of heritage should be the coherent and whole consideration from the heritage's internal environment to the external environment, which covered the heritage architecture, art activity, wide nature, and regional attraction.

Satisfying tourists must be compatible with heritage sustainable management [65]. The dimensions of the recreational benefits and long-term values reflect intrinsic motivation. The recreational benefits dimension indicated that heritage tourism should provide a novel and fun experience that allows visitors to share travel experiences and that promotes interaction with family and friends. Interactive experiences are often an important need to induce home-like experiences and place attachment [33,66]. Meeting tourists' expectations is an important indicator for assessing heritage tourism potential, such as offering fun experiences [67]. The dimension of long-term values indicates that heritage tourism should provide a sense of achievement, promote a good life, and improve health. The second-order structure indicated that the internal to external heritage environment correlated with recreational benefits and long-term values, including heritage architecture, art activity, wide nature, and regional attraction. For example, green spaces often produce opportunities for social interaction with friends and families, resulting in a positive effect on health [68]. Nature also positively produces fun activity and experience [15] and yields home-like experiences and place attachment [66]. Biophilia and cultural elements can induce emotional attachment to affect health [3]. Connecting historical and natural environments also enhances the purpose of leisure and recreation [69]. This kind of restorative experience also contributes to tourists' positive emotions, life satisfaction, and intention to revisit [70]. These benefits and values reflect not only intrinsic motivation, but also the need for biophilia.

## Meaning of the second-order factor structure

The results showed that the second-order model had adequate values of goodness-of-fit indices, convergent validity, and discriminant validity. The high target coefficients of the two models also constitute evidence of a second-order factor. The six correlated dimensions were integrated into a common and meaningful second-order factor. This second-order factor can be called biophilia heritage tourism motivation because the six dimensions are clearly connected to biophilia motivation and elements. The second-order factor is completely latent, unobservable, and not measurable because there are no indicators in the second-order factor [34]. Indeed, the biophilia heritage tourism motivation presented as an abstract concept should be explained and observed by the main and sub-structures.

Interestingly, the heritage architecture dimension produced a low factor loading in the second-order model, suggesting that the heritage architecture dimension may not be a core dimension for assessing biophilia heritage tourism motivation. However, the heritage architecture dimension still had the highest importance level among respondents. The second-order model also produced acceptable convergent validity. Thus, this study retains the heritage architecture dimension. The other five dimensions would represent biophilia tourism motivation if the study did not consider the heritage architecture dimension. The essence of heritage tourism may cover the features of biophilia tourism. The essence of heritage tourism and general tourism may be biophilia. Further studies should examine the concept of biophilia in general tourism.

The second-order model produces both practical and academic applications. First, sustainable heritage preservation should consider the biophilia heritage tourism motivation, including heritage architecture, art activity, wide nature, regional attraction, recreational benefits, and long-term values. These six dimensions are crucial principles for achieving sustainable heritage development to satisfy participants' needs. Further studies should construct the indicators with weights to establish an assessment by the analytic hierarchy process. Second, assessing heritage sites' tourism potential is an important issue for attracting tourists [11,71]. Heritage managers and the government can use these six dimensions to check the development potential of adaptive heritage reuse. The second-order model's confirmation allows the use of an overall score of 15 items to assess sustainable heritage development's tourism potential and determine heritage reuse selection. For example, heritage sites and locations affect the potential for wide nature and regional attractions [72].

## Limitations and future research

Some limitations should be indicated. First, culture is a potential limitation of this study. Different cultures produce different motivations and values [73] that may affect the motivation for heritage tourism. Although cultural background may influence tourism motivation, biophilia motivation may be stable from a similar evolutionary tendency. For example, nature is a common biophilia motivation in multiple types of tourism. Further studies should examine the biophilia motivations of heritage tourism across cultures to increase the external validity of the factor structure. Second, this study did not directly evidence the biophilia heritage motivation, although this study attempted to link the relationship between heritage and biophilia. Biophilia is an abstract concept that is hard to evidence in evolutionary terms. Indirect evidence is a problem in biophilia studies [34].

## Conclusions

This study provides a confirmative framework for the motivation of biophilic heritage tourism. The heritage tourism motivation framework consisted of a second-order six-factor structure with high validity and reliability, following the CFA process, including heritage architecture, art activity, wide nature, regional attraction, recreational benefits, and long-term values. In the heritage architecture dimension, heritage architecture, the beautification of heritage architecture, and heritage architecture style should be considered for protection or representation. In the art activity dimension, heritage sites should hold dynamic and static art activities and holiday markets to promote their critical intangible cultural elements and increase tourism motivation. Natural elements and natural and wide landscapes are critical elements to improve a heritage structure's environment. The regional attractions were also found to be important elements for heritage tourism, including historical streets, cultural heritage sites, and recreational attractions. The novel and fun experience, sharing of travel experiences, promoting interaction with family and friends, feeling a sense of achievement, promoting a good life, and improving health were observed as the recreational benefits and long-term values associated with heritage tourism, reflecting the intrinsic motivation for tourists to visit these sites and satisfy their biophilic requirements. These six dimensions reflected the biophilic requirements and thereby provided a biophilic planning principle to achieve sustainable heritage management to satisfy participants' needs. The second-order model's confirmation allows the use of an overall score to assess the potential of sustainable heritage development and determine the selection of heritage reuse. Heritage tourism with the above six dimensions will have a higher potential for attracting tourists.

## Author Contributions

**Conceptualization:** Hung-Ming Tu.

**Data curation:** Hung-Ming Tu.

**Formal analysis:** Hung-Ming Tu.

**Investigation:** Hung-Ming Tu.

**Methodology:** Hung-Ming Tu.

**Project administration:** Hung-Ming Tu.

**Supervision:** Hung-Ming Tu.

**Writing – original draft:** Hung-Ming Tu.

**Writing – review & editing:** Hung-Ming Tu.

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
