## [Decision Letter · Decision Letter 0]

26 Jan 2022

PONE-D-21-25347Confirmative Biophilic Framework for Heritage ManagementPLOS ONE

Dear Dr. Hung-Ming Tu,

Thank you for submitting your manuscript to PLOS ONE. After careful consideration, we feel that it has merit but does not fully meet PLOS ONE’s publication criteria as it currently stands. Therefore, we invite you to submit a revised version of the manuscript that addresses the points raised during the review process.

We look forward to receiving your revised manuscript.

Kind regards,

Carlos Alberto Zúniga-González, Ph.D

Academic Editor

PLOS ONE

https://journals.plos.org/plosone/s/file?id=ba62/PLOSOne_formatting_sample_title_authors_affiliations.pdf”.

“The author is sincerely grateful to the Ministry of Science and Technology (MOST) in Taiwan for its full support (Project No. MOST 107-2410-H-005-028-MY3).”

 “Hung Ming Tu

MOST 107-2410-H-005-028-MY3

Ministry of Science and Technology (MOST) in Taiwan

https://www.most.gov.tw/

Additional Editor Comments:

Dear author, your manuscript is interesting and useful for the tourism economy. I suggest you attend to the comments of the reviewers in order to improve the quality of your manuscript.

Reviewers' comments:

Reviewer's Responses to Questions

**Comments to the Author**

1. Is the manuscript technically sound, and do the data support the conclusions?

Reviewer #1: Yes

Reviewer #2: Yes

2. Has the statistical analysis been performed appropriately and rigorously? 

Reviewer #1: Yes

Reviewer #2: Yes

3. Have the authors made all data underlying the findings in their manuscript fully available?

Reviewer #1: Yes

Reviewer #2: Yes

4. Is the manuscript presented in an intelligible fashion and written in standard English?

Reviewer #1: Yes

Reviewer #2: Yes

5. Review Comments to the Author

Reviewer #1: A study of how biophilic design influences heritage architecture and sites. Extremely important for the tourist industry and local economy. Questionnaires created by the author gave statistical data of preferences. Please indicate at the beginning of this paper where this study was done! A more serious omission is the author's lack of literature knowledge. He/she focuses upon natural elements, and does not delve into the biophilic design elements due to the materials and building details and geometry, which is the most relevant to heritage structures because those are higher there than in any recent buildings. This conceptual gap needs to be filled in before the paper is ready for publication, otherwise it is misleading. Other than this additional descriptive explanation, the technical analysis of data is fine. Finally, the Conclusion needs to be re-written and expanded. A reader cannot gain an understanding of the general problem or the result from reading the present vague and brief text. This is an important topic, and the author needs to help the reader understand what this is about -- from the Conclusion. Some relevant references:

https://www.mdpi.com/2071-1050/13/2/823

http://www.biourbanism.org/the-biophilic-healing-index-predicts-effects-of-the-built-environment-on-our-wellbeing/

https://www.terrapinbrightgreen.com/wp-content/uploads/2015/10/Biophilia-Healing-Environments-Salingaros-p.pdf

Reviewer #2: The document is well written, it presents solids in the discussion. For the author, I suggest that he reinforce the discussion with more sources to support the results. Be very careful with repeated phrases in the document. As far as the statistical analysis is concerned, the document presents a good analysis.

6. PLOS authors have the option to publish the peer review history of their article (what does this mean?). If published, this will include your full peer review and any attached files.

Reviewer #1: No

Reviewer #2: No

---

## [Author Response · Author response to Decision Letter 0]

9 Mar 2022

Submission ID PONE-D-21-25347

Confirmative Biophilic Framework for Heritage Management 

Additional Editor Comments:

Dear author, your manuscript is interesting and useful for the tourism economy. I suggest you attend to the comments of the reviewers in order to improve the quality of your manuscript.

【Our responses】: 

We thank you very much for giving us the opportunity to revise our manuscript. We are grateful for the reviewer’s comments and suggestions, which significantly contributed to improving the quality of this paper.

Comments to the Author

1. Is the manuscript technically sound, and do the data support the conclusions?

Reviewer #1: Yes

Reviewer #2: Yes

2. Has the statistical analysis been performed appropriately and rigorously?

Reviewer #1: Yes

Reviewer #2: Yes

3. Have the authors made all data underlying the findings in their manuscript fully available?

Reviewer #1: Yes

Reviewer #2: Yes

4. Is the manuscript presented in an intelligible fashion and written in standard English?

Reviewer #1: Yes

Reviewer #2: Yes

【Our responses】: 

We thank you very much for giving us the opportunity to revise our manuscript. We are grateful for the reviewer’s comments and suggestions, which significantly contributed to improving the quality of this paper.

 

5. Review Comments to the Author 

Comments from the Reviewer 1

A study of how biophilic design influences heritage architecture and sites. Extremely important for the tourist industry and local economy. Questionnaires created by the author gave statistical data of preferences. Please indicate at the beginning of this paper where this study was done! 

A more serious omission is the author's lack of literature knowledge. He/she focuses upon natural elements, and does not delve into the biophilic design elements due to the materials and building details and geometry, which is the most relevant to heritage structures because those are higher there than in any recent buildings. This conceptual gap needs to be filled in before the paper is ready for publication, otherwise it is misleading. Other than this additional descriptive explanation, the technical analysis of data is fine. Finally, the Conclusion needs to be re-written and expanded. A reader cannot gain an understanding of the general problem or the result from reading the present vague and brief text. This is an important topic, and the author needs to help the reader understand what this is about -- from the Conclusion. Some relevant references:

https://www.mdpi.com/2071-1050/13/2/823

http://www.biourbanism.org/the-biophilic-healing-index-predicts-effects-of-the-built-environment-on-our-wellbeing/

https://www.terrapinbrightgreen.com/wp-content/uploads/2015/10/Biophilia-Healing-Environments-Salingaros-p.pdf

【Our responses】: 

Thank you for your comments and suggestions, which enabled us to enhance the quality of this paper. We added the references to discuss building details and re-written the conclusion.

The natural light, bright colors, rich borders, frames, moldings, ornaments, natural materials, balanced curves, water, plants, and non-threatening animals are important considerations for a biophilic design in the built environment [23-25]. In the heritage architecture dimension, natural fractal patterns may play an important role in visual feeling. A natural fractal pattern is a special type of complexity that produces a positive effect on aesthetic experience, preference, emotion, and stress reduction [26]. This is because our ancestors may have identified the non-fractal patterns of animals to avoid dangerous situations and thereby adopted a sensitivity toward natural fractal patterns [26]. In a built environment, the architecture, flooring design, windows, and decorations also use natural fractal patterns to promote an aesthetic experience and preference [26]. For example, the representations of plants, animals, and people in the built environment through photographs, paintings, or sculptures contribute to biophilia [23,24]. The designs of several classical architecture ornaments are derived from leaves, flowers, and animal skins [27]. However, modern architecture often portrays minimalism to erase biophilic features [23,24], which may lead to a reduction in an individual’s positive reflection on preference and emotion. (Lines 108-121).

Vernacular architecture often presents the richness and variety of biophilic features to produce a healing effect due to the complex geometry of the neurobiological system's preferences [23]. (Lines 122-123).

The elements of architecture and its beautification are consistent with the considerations of biophilic design. Natural fractal patterns and representations of plants, leaves, flowers, animals, and people should be considered to promote biophilic experiences and health [26]. (Lines 309-312)

Traditional Japanese gardens present small-scale biophilic interventions in people’s doorways [27]. (Lines 315-316)

Considering their effectiveness, cost, and benefit-cost ratio, Xue et al. indicated that natural window views, natural ventilation, and natural landscape promotion with minimal management are the three most essential strategies in a built environment from the perspective of stakeholders [63]. This study also supported the proposition that heritage architecture sites should consider natural window views and natural landscape promotion. One study indicated that the natural environment should not obstruct the view of historical structures at the cost of promoting landscape preferences [64], implying that the aesthetic and natural view of historical structures should be an important consideration while undertaking heritage greening activities. Interestingly, natural decoration and ornamentation are not considered cost-effective or favored in the general built environment by stakeholders [63]. This study suggested that heritage decoration and ornamentation are important considerations for heritage tourism. Therefore, the decoration of natural fractal patterns should be considered for heritage protection or tourism. (Lines 347-358)

This kind of restorative experience also contributes to tourists’ positive emotions, life satisfaction, and intention to revisit [70]. (Lines 383-384)

This study provides a confirmative framework for the motivation of biophilic heritage tourism. The heritage tourism motivation framework consisted of a second-order six-factor structure with high validity and reliability, following the CFA process, including heritage architecture, art activity, wide nature, regional attraction, recreational benefits, and long-term values. In the heritage architecture dimension, heritage architecture, the beautification of heritage architecture, and heritage architecture style should be considered for protection or representation. In the art activity dimension, heritage sites should hold dynamic and static art activities and holiday markets to promote their critical intangible cultural elements and increase tourism motivation. Natural elements and natural and wide landscapes are critical elements to improve a heritage structure’s environment. The regional attractions were also found to be important elements for heritage tourism, including historical streets, cultural heritage sites, and recreational attractions. The novel and fun experience, sharing of travel experiences, promoting interaction with family and friends, feeling a sense of achievement, promoting a good life, and improving health were observed as the recreational benefits and long-term values associated with heritage tourism, reflecting the intrinsic motivation for tourists to visit these sites and satisfy their biophilic requirements. These six dimensions reflected the biophilic requirements and thereby provided a biophilic planning principle to achieve sustainable heritage management to satisfy participants’ needs. The second-order model’s confirmation allows the use of an overall score to assess the potential of sustainable heritage development and determine the selection of heritage reuse. Heritage tourism with the above six dimensions will have a higher potential for attracting tourists. (Lines 428-447)

References:

[23] Salingaros NA. Biophilia and healing environments: Healthy principles for designing the built world. Metropolis and Terrapin Bright Green. LLC; 2015.

[24] Salingaros NA. The biophilic healing index predicts effects of the built environment on our wellbeing. J. Biourbanism. 2019;8(1):13-34.

[25] Gillis K, Gatersleben B. A review of psychological literature on the health and wellbeing benefits of biophilic design. Buildings. 2015;5(3):948-963. doi:10.3390/buildings5030948

[26] Taylor RP. The potential of biophilic fractal designs to promote health and performance: A review of experiments and applications. Sustainability. 2021;13(2):823. doi:10.3390/su13020823

[27] Ryan CO, Browning WD. Biophilic Design. In: Loftness V. (ed.) Sustainable Built Environments. Encyclopedia of Sustainability Science and Technology Series. Springer, New York, NY; 2020. doi:10.1007/978-1-0716-0684-1_1034

[63] Xue F, Gou Z, Lau SSY, Lau SK, Chung KH, Zhang J. From biophilic design to biophilic urbanism: Stakeholders’ perspectives. J. Clean Prod. 2019;211:1444-1452. doi:10.1016/j.jclepro.2018.11.277

[64] Pardela Ł, Lis A, Iwankowski P, Wilkaniec A, Theile M. The importance of seeking a win-win solution in shaping the vegetation of military heritage landscapes: The role of legibility, naturalness and user preference. Urban Landsc. Plan. 2022;221:104377. doi:10.1016/j.landurbplan.2022.104377

[70] Backman SJ, Huang YC, Chen CC, Lee HY, Cheng JS. Engaging with restorative environments in wellness tourism. Curr. Issues Tour. 2022;1-18. doi:10.1080/13683500.2022.2039100

 

Comments from the Reviewer 2

The document is well written, it presents solids in the discussion. For the author, I suggest that he reinforce the discussion with more sources to support the results. Be very careful with repeated phrases in the document. As far as the statistical analysis is concerned, the document presents a good analysis.

【Our responses】:

Thank you for your comments and suggestions, which enabled us to enhance the quality of this paper. We added the references to reinforce discussion section.

The natural light, bright colors, rich borders, frames, moldings, ornaments, natural materials, balanced curves, water, plants, and non-threatening animals are important considerations for a biophilic design in the built environment [23-25]. In the heritage architecture dimension, natural fractal patterns may play an important role in visual feeling. A natural fractal pattern is a special type of complexity that produces a positive effect on aesthetic experience, preference, emotion, and stress reduction [26]. This is because our ancestors may have identified the non-fractal patterns of animals to avoid dangerous situations and thereby adopted a sensitivity toward natural fractal patterns [26]. In a built environment, the architecture, flooring design, windows, and decorations also use natural fractal patterns to promote an aesthetic experience and preference [26]. For example, the representations of plants, animals, and people in the built environment through photographs, paintings, or sculptures contribute to biophilia [23,24]. The designs of several classical architecture ornaments are derived from leaves, flowers, and animal skins [27]. However, modern architecture often portrays minimalism to erase biophilic features [23,24], which may lead to a reduction in an individual’s positive reflection on preference and emotion. (Lines 108-121).

Vernacular architecture often presents the richness and variety of biophilic features to produce a healing effect due to the complex geometry of the neurobiological system's preferences [23]. (Lines 122-123).

The elements of architecture and its beautification are consistent with the considerations of biophilic design. Natural fractal patterns and representations of plants, leaves, flowers, animals, and people should be considered to promote biophilic experiences and health [26]. (Lines 309-312)

Traditional Japanese gardens present small-scale biophilic interventions in people’s doorways [27]. (Lines 315-316)

Considering their effectiveness, cost, and benefit-cost ratio, Xue et al. indicated that natural window views, natural ventilation, and natural landscape promotion with minimal management are the three most essential strategies in a built environment from the perspective of stakeholders [63]. This study also supported the proposition that heritage architecture sites should consider natural window views and natural landscape promotion. One study indicated that the natural environment should not obstruct the view of historical structures at the cost of promoting landscape preferences [64], implying that the aesthetic and natural view of historical structures should be an important consideration while undertaking heritage greening activities. Interestingly, natural decoration and ornamentation are not considered cost-effective or favored in the general built environment by stakeholders [63]. This study suggested that heritage decoration and ornamentation are important considerations for heritage tourism. Therefore, the decoration of natural fractal patterns should be considered for heritage protection or tourism. (Lines 347-358)

This kind of restorative experience also contributes to tourists’ positive emotions, life satisfaction, and intention to revisit [70]. (Lines 383-384)

References:

[23] Salingaros NA. Biophilia and healing environments: Healthy principles for designing the built world. Metropolis and Terrapin Bright Green. LLC; 2015.

[24] Salingaros NA. The biophilic healing index predicts effects of the built environment on our wellbeing. J. Biourbanism. 2019;8(1):13-34.

[25] Gillis K, Gatersleben B. A review of psychological literature on the health and wellbeing benefits of biophilic design. Buildings. 2015;5(3):948-963. doi:10.3390/buildings5030948

[26] Taylor RP. The potential of biophilic fractal designs to promote health and performance: A review of experiments and applications. Sustainability. 2021;13(2):823. doi:10.3390/su13020823

[27] Ryan CO, Browning WD. Biophilic Design. In: Loftness V. (ed.) Sustainable Built Environments. Encyclopedia of Sustainability Science and Technology Series. Springer, New York, NY; 2020. doi:10.1007/978-1-0716-0684-1_1034

[63] Xue F, Gou Z, Lau SSY, Lau SK, Chung KH, Zhang J. From biophilic design to biophilic urbanism: Stakeholders’ perspectives. J. Clean Prod. 2019;211:1444-1452. doi:10.1016/j.jclepro.2018.11.277

[64] Pardela Ł, Lis A, Iwankowski P, Wilkaniec A, Theile M. The importance of seeking a win-win solution in shaping the vegetation of military heritage landscapes: The role of legibility, naturalness and user preference. Urban Landsc. Plan. 2022;221:104377. doi:10.1016/j.landurbplan.2022.104377

[70] Backman SJ, Huang YC, Chen CC, Lee HY, Cheng JS. Engaging with restorative environments in wellness tourism. Curr. Issues Tour. 2022;1-18. doi:10.1080/13683500.2022.20391007

---

## [Editor Report · Decision Letter 1]

15 Mar 2022

Confirmative Biophilic Framework for Heritage Management

PONE-D-21-25347R1

Dear Dr. Hung-Ming Tu,

We’re pleased to inform you that your manuscript has been judged scientifically suitable for publication and will be formally accepted for publication once it meets all outstanding technical requirements.

Kind regards,

Carlos Alberto Zúniga-González, Ph.D

Academic Editor

PLOS ONE

Additional Editor Comments (optional):

I have reviewed that you have improved the quality of your manuscript by incorporating the comments of the reviewers. My sincerest congratulations.
---

## [Editor Report · Acceptance letter]

21 Mar 2022

PONE-D-21-25347R1 

Confirmative Biophilic Framework for Heritage Management 

Dear Dr. Tu:

I'm pleased to inform you that your manuscript has been deemed suitable for publication in PLOS ONE. Congratulations! Your manuscript is now with our production department. 

Kind regards, 

on behalf of

Dr. Prof. Carlos Alberto Zúniga-González 

Academic Editor

PLOS ONE